# Induction of Extracellular Hydroxyl Radicals Production in the White-Rot Fungus *Pleurotus eryngii* for Dyes Degradation: An Advanced Bio-oxidation Process

**DOI:** 10.3390/jof10010052

**Published:** 2024-01-07

**Authors:** Ana Belén García-Martín, Juana Rodríguez, José Manuel Molina-Guijarro, Carmen Fajardo, Gabriela Domínguez, Manuel Hernández, Francisco Guillén

**Affiliations:** Department of Biomedicine and Biotechnology, Universidad de Alcalá, 28805 Alcalá de Henares, Spain; belen.garciamartin@educa.madrid.org (A.B.G.-M.); juana.rodriguez@uah.es (J.R.); josemanuel.molina@uah.es (J.M.M.-G.); carmen.fajardo@uah.es (C.F.); gabriela.dominguez@uah.es (G.D.); manuel.hernandez@uah.es (M.H.)

**Keywords:** *Pleurotus eryngii*, white-rot fungi, advanced bio-oxidation process (ABOP), quinone redox cycling, hydroxyl radicals, dyes degradation, advanced oxidation process (AOP)

## Abstract

Among pollution remediation technologies, advanced oxidation processes (AOPs) are genuinely efficient since they are based on the production of strong, non-selective oxidants, mainly hydroxyl radicals (·OH), by a set of physicochemical methods. The biological counterparts of AOPs, which may be referred to as advanced bio-oxidation processes (ABOPs), have begun to be investigated since the mechanisms of induction of ·OH production in fungi are known. To contribute to the development of ABOPs, advanced oxidation of a wide number of dyes by the white-rot fungus *Pleurotus eryngii*, via a quinone redox cycling (QRC) process based on Fenton’s reagent formation, has been described for the first time. The fungus was incubated with 2,6-dimethoxy-1,4-benzoquinone (DBQ) and Fe^3+^-oxalate, with and without Mn^2+^, leading to different ·OH production rates, around twice higher with Mn^2+^. Thanks to this process, the degradative capacity of the fungus increased, not only oxidising dyes it was not otherwise able to, but also increasing the decolorization rate of 20 dyes by more than 7 times in Mn^2+^ incubations. In terms of process efficacy, it is noteworthy that with Mn^2+^ the degradation of the dyes reached values of 90–100% in 2–4 h, which are like those described in some AOPs based on the Fenton reaction.

## 1. Introduction

The growing demand from society for the design of strategies for the decontamination of ecosystems, which has resulted in increasingly stringent governmental regulations, has in recent years encouraged the development of new technologies for their treatment. These technologies are based on physicochemical and biological processes. Among them, the so-called advanced oxidation processes (AOPs) [1] and procedures involving white-rot fungi (WRF) and their enzymes [2] are currently attracting the most attention among researchers and technologists. WRF, a famous group of degraders in nature, possess the capability to degrade both lignin and cellulose biopolymers within lignocellulose biomass. Concurrently, they exhibit unique oxidative and extracellular ligninolytic systems characterized by low substrate specificity, enabling them to transform or degrade diverse environmental contaminants [2,3]. The heightened interest in AOPs and WRF stems from two key attributes desired in degradative agents: high reactivity and low selectivity. These qualities are essential for attacking persistent contaminants and dealing with a broad spectrum and variety of such substances. Both the oxidants produced in AOPs and the enzymes and oxidants generated by WRF exhibit these characteristics, making them highly effective in the remediation of recalcitrant contaminants. 

The degradation of pollutants based on the action of highly reactive species such as, primarily but not exclusively, hydroxyl radicals (·OH) is referred to as advanced oxidation. In 1987, Glaze et al. used time the term AOPs for the first to define “near ambient temperature and pressure water treatment processes which involve the generation of hydroxyl radicals in sufficient quantity to effect water purification” [4]. Since then, a broad number of technologies generating ·OH for pollutant degradation have been described and developed, mainly for water and wastewater treatment [5]. Most of these technologies use a combination of chemical oxidants (e.g., H_2_O_2_, O_3_) with catalysts (e.g., transition metal ions) and auxiliary energy sources (e.g., ultraviolet and visible radiations, electric current, g-radiation, ultrasound, etc.) to generate ·OH. Examples of AOPs include O_3_/H_2_O_2_, Fenton’s reagent (Fe^2+^/H_2_O_2_), photo-Fenton (UV/Fe^2+^/H_2_O_2_), electro-Fenton, UV/H_2_O_2_, UV/O_3_, heterogeneous photocatalysis (TiO_2_/UV), γ-radiolysis, etc. [6,7]. AOPs have shown to be highly effective with a huge number of pollutants and wastewater from several industries: pesticides [8], benzene, toluene, ethylbenzene and xylene isomers (BTEX) [9], surfactants [10], dyes [11], chlorinated aromatic hydrocarbons [12], food industry [13], olive mill [14], petroleum refinery [15], etc. The effectiveness of AOPs for the degradation of a huge number of pollutants is due to the high reactivity and non-selectivity of ·OH radicals. They have a reduction potential of 2.8 V and react with most organic molecules at rate constants in the order of 10^8^–10^11^ M^−1^ s^−1^. 

Compared with AOP, biological procedures used for pollutant degradation are much more selective as they are based on the action of enzymes, which in general show a high degree of substrate specificity. However, there exists a group of microorganisms, the WRF, which has been proven to degrade a high number and variety of pollutants [16,17]. This ability is mainly due to the secretion of an enzyme system which shows high reactivity and low substrate specificity and whose natural function is the degradation of lignin [18,19]. The composition of the ligninolytic system varies among species, but it is generally integrated by a combination of laccase and several types of peroxidases, i.e., lignin peroxidase (LiP), manganese peroxidase (MnP), and versatile peroxidase (VP). The H_2_O_2_ required for the activity of peroxidases is provided by extracellular oxidases, such as aryl-alcohol oxidase (AAO) and glyoxal oxidase [19]. Ligninolytic enzymes catalyse the one electron oxidation of substrates, producing free radicals that undergo a variety of spontaneous reactions leading to their degradation [20]. In addition to acting directly, the ligninolytic system can bring about lignin and pollutant degradation through the generation of low molecular weight chemical oxidants, including Mn^3+^ and free radicals from some fungal metabolites and lignin depolymerisation products which are termed enzyme mediators [21,22]. This mediated oxidation mechanism, that resembles AOPs in the generation of reactive radicals as the degradative agents, expands the range of pollutants susceptible to degradation by the ligninolytic system since substrate specificity stops being a limitation [23]. 

WRF also present the ability to produce extracellular ·OH [24]. These radicals, along with Mn^3+^ and enzyme mediator radicals, are the low molecular weight degradative agents participating in the initial attack of lignocellulose when ligninolytic enzymes cannot penetrate plant cell walls [25]. Although ·OH are the strongest oxidants found in cultures of WRF, studies on both their involvement in and their exploitation for pollutants degradation are scarce. To date, five potential mechanisms of extracellular ·OH production in WRF, one chemical [26] and four enzymatic [27,28,29,30], have been described. All of them are based on the production of ·OH by the Fenton reaction (Fe^2+^ + H_2_O_2_ → F^3+^ + ·OH + OH^−^) and have been inferred from the results obtained in in vitro reactions carried out with different combinations of fungal metabolites, lignin depolymerisation products, iron complexes, and one of the different enzymes involved in lignocellulose degradation. In the chemical mechanism, an extracellular fungal glycopeptide has been shown to reduce F^3+^ to Fe^2+^ and O_2_ to H_2_O_2_ via superoxide anion radical (O_2_·^−^) [26]. Enzymatic mechanisms include two enzymes participating in lignin and cellulose degradation, i.e., cellobiose dehydrogenase (CDH) and lytic polysaccharide monooxygenase (LPMO), and the ligninolytic enzymes LiP and laccase. Among them, only CDH is able to directly catalyse the formation of Fenton’s reagent, using cellodextrines as electron donors [29]. The ligninolytic enzymes and LPMO, however, act as an indirect source of ·OH through the generation of Fe^3+^ and O_2_ reductants, such as formate radicals (CO_2_·^−^) and semiquinone radicals (Q·^−^). The first-time it was demonstrated that a ligninolytic enzyme could be involved in ·OH production, oxalate was used to generate CO_2_·^−^ in LiP reactions mediated by veratryl alcohol [27]. The proposed mechanism consisted of the following cascade of reactions: (i) production of veratryl alcohol cation radical (Valc·^+^) by LiP; (ii) oxidation of oxalate to CO_2_·^−^ by Valc·^+^; (iii) reduction of O_2_ to O_2_·^−^ by CO_2_·^−^; (iv) H_2_O_2_ production by O_2_·^−^ dismutation (O_2_·^−^ + HO_2_· + H^+^ → O_2_ + H_2_O_2_); and (v) superoxide-driven Fenton reaction (Haber-Weiss reaction), where Fe^3+^-EDTA is reduced to Fe^2+^-EDTA by O_2_·^−^. The ·OH production mechanism assisted by Q·^−^ was inferred from the oxidation by laccase of two lignin-derived hydroquinones (QH_2_), i.e., 2-methoxy-1,4-benzohydroquinone (MBQH_2_) and 2,6-dimethoxy-1,4-benzohydroquinone (DBQH_2_), in the presence of Fe^3+^-EDTA [28]. The sequence of reactions leading to ·OH generation were the following: (i) oxidation of QH_2_ to Q·^−^ radicals by laccase; (ii) Q·^−^ autoxidation catalysed by Fe^3+^-EDTA, in which Fe^2+^-EDTA and O_2_·^−^ are generated (Q·^−^ + Fe^3+^-EDTA → Q + Fe^2+^-EDTA; and Fe^2+^-EDTA + O_2_ ⇌ Fe^3+^-EDTA + O_2_·^−^); (iii) H_2_O_2_ production by O_2_·^−^ dismutation; and (iv) Fenton reaction. In this case, ·OH radicals were generated by a semiquinone-driven Fenton reaction because Q·^−^ radicals were the main agents accomplishing Fe^3+^ reduction. Finally, the more recently discovered LPMO, which was first demonstrated to be involved in cellulose degradation, has been shown to be able to oxidise DBQH_2_, leading to a semiquinone-driven Fenton reaction, as described for laccase. LPMO is also able of directly reducing O_2_ to H_2_O_2_ [30]. 

Based on the mechanism involving laccase and hydroquinones [28,31], a strategy for the induction of extracellular ·OH production was first demonstrated in in vivo experiments carried out with *Pleurotus eryngii*, which was grown under conditions producing either laccase and VP or only laccase. *P. eryngii* is a well-known grassland-inhabiting fungus showing high ability to colonize non-woody lignocellulosic materials by the production of a wide array of oxidative and hydrolytic enzymes [32]. The above ·OH induction strategy was subsequently confirmed in other well-studied WRF, i.e., *Trametes versicolor*, *Phanerochaete chrysosporium*, *Bjerkandera adusta*, *Phlebia radiata*, and *Pycnoporus cinnabarinus*, each one producing different pools of ligninolytic enzymes [33]. The strategy consists of the incubation of fungi with a lignin-derived quinone (Q) and conveniently chelated ferric ion. Under these conditions, ·OH radicals are produced through a quinone redox cycling (QRC) mechanism. This mechanism is catalysed by an intracellular quinone reductase (QR), which converts Q into QH_2_ (that is secreted to the extracellular medium), and any of the ligninolytic enzymes, which oxidises QH_2_ to Q·^−^, triggering the formation of Fenton’s reagent and ·OH radicals. In other words, it was demonstrated that extracellular ·OH were produced by a semiquinone-driven Fenton mechanism in which quinones act as electron carriers between intracellular electron equivalents used by QR (NADH or NADPH) and extracellular O_2_. This ·OH-producing mechanism provided useful information for the design of experiments aimed at demonstrating the involvement of these radicals in the oxidation of pollutants and improving the already high capability of WRF. For instance, the use of this strategy to induce ·OH production in *P. eryngii* provided the first proof of the advanced oxidation of two pollutants (phenol and the dye reactive black 5) by a WRF [34]. 

By analogy with AOPs, an advanced bio-oxidation process (ABOP) can be defined as a biological procedure that promotes and takes advantage of the production of highly reactive and non-selective oxidants, such as ·OH, for pollutant degradation. As in AOPs based on the Fenton reaction (Fenton-like, photo-Fenton, electro-Fenton, etc.) [35], QRC mechanisms allow for quantitative control over the levels of ·OH produced. This control may depend either on the type of quinone and iron chelator used, on the metal ions and metabolites that increase the levels of hydrogen peroxide and on the rate of the quinone redox cycle [28,31,33,34]. First, among the quinones representing the three lignin units, i.e., 1,4-benzoquinone (BQ, 4-hydroxyphenyl unit), 2-methoxy-1,4-benzoquinone (MBQ, guaiacyl unit), and 2,6-dimethoxy-1,4 benzoquinone (DBQ, syringyl unit), the latter was the one rendering the highest ·OH levels. The reasons are that the ligninolytic enzymes present higher activity on DBQH_2_ and the semiquinone radical produced is oxidised at a higher rate by both O_2_ and Fe^3+^ complexes. Second, among EDTA and oxalate used to complex Fe^3+^, Fe^3+^-oxalate had the ability to oxidise not only DBQ·^−^ but also DBQH_2_ (hydroquinone-driven Fenton mechanism). This way, the rate of the redox cycle is enhanced, provided that hydroquinone oxidation is its limiting reaction, which is usual when ligninolytic enzyme levels are relatively low. Third, besides Fe^3+^, a second key cation playing an important role in lignin degradation, i.e., Mn^2+^, was also relevant in QRC since it reduces the O_2_·^−^ derived from semiquinone radicals autoxidation (Mn^2+^ + O_2_·^−^ + 2H^+^ → Mn^3+^ + H_2_O_2_), enhancing H_2_O_2_ levels. In addition, the resulting Mn^3+^ cation propagates hydroquinones oxidation (Mn^3+^ + DBQH_2_ → Mn^2+^ + DBQ·^−^) and, thus, the redox cycle rate. Finally, substrates of oxidases, such as the AAO produced by *P. eryngii*, increase H_2_O_2_ levels and, thus, ·OH production, via a redox cycling mechanism involving an intracellular aldehyde reductase and the extracellular AAO [19,34]. A scheme of ·OH production by *P. eryngii*, via QRC, showing the role that Fe^3+^-oxalate, Mn^2+^, and aromatic aldehydes play in the process was included in the preceding study of Gómez-Toribio et al. [34]. 

Based on these findings, several studies have addressed the use of this fungal-mediated ABOP for the degradation of different types of pollutants [36]. To further explore this degradation strategy and to strengthen the use of the term ABOP, the aim of the present study was to demonstrate that the induction in *P. eryngii* of the extracellular production of ·OH, via QRC, allows for the fungus to degrade the largest number and variety of contaminant compounds used so far in an ABOP due to the high oxidation power and low selectivity of these radicals. For this purpose, we selected 27 dyes of the following structural types: anthraquinone, azo, diazo, heterocycle, indigo, phthalocyanine, and triarylmethane.

## 2. Materials and Methods

### 2.1. Dyes, Other Chemicals, and Enzymes

The twenty-seven dyes used in this study were obtained from Sigma-Aldrich. According to their structure, they have been grouped into 7 types (the dye content is shown in brackets). Anthraquinone type: Acid Black 48 (40%), Acid Blue 45 (50%), Acid Green 25 (75%), and Reactive Blue 19 (~50%). Azo type: Acid Red 88 (78%), Acid Yellow 17 (60%), Chromotrope 2R (75%), Crocein Orange G (70%), Methyl Orange (85%), New Coccine (75%), Orange II (~85%), Tartrazine (~60%), and Tropaeolin O (65%). Diazo type: Acid Black 24 (50%), Acid Blue 113 (50%), Acid Orange 63 (45%), Ponceau SS (80%), and Reactive Black 5 (55%). Heterocyclic type: Azure B (96%) and Neutral Red (~90%). Indigo type: Indigo Carmine (90%). Phthalocyanine type: Iron(III)phthalocyanine (~90%); Triarylmethane type: Brilliant Green (~90%), Bromophenol Blue (90%), Cresol Red (90%), Crystal Violet (~90%), and Methyl Blue (~60%). The chemical structure and maximal absorption wavelength (λ_max_) of each dye are shown in Appendix A.

2,2′-azino-bis(3-ethylbenzothiazoline-6-sulfonic acid) (ABTS), oxalic acid, 2-thiobarbituric acid (TBA), ethylenediaminetretaacetic acid (EDTA), and 2-deoxyribose were purchased from Sigma. 1,4-benzoquinone (BQ), 2,6-dimethoxy-1,4-benzoquinone (DBQ), 2,6-dimethoxyphenol (DMP), and FeCl_3_ were supplied by Aldrich. DBQH_2_ was prepared by DBQ reduction with sodium borohydride [37] and kept at 4 °C under an argon atmosphere to avoid oxidation. All other chemicals used were of analytical grade. 

A preparation of partially purified *P. eryngii* laccase, containing a mixture of the two isoenzymes produced by the fungus, was obtained as previously described [38].

### 2.2. Organism and Culture Conditions

*P. eryngii* IJFM A169 (Fungal Culture Collection of the Centro de Investigaciones Biológicas = ATCC 90787 and CBS 613.91) was maintained at 4 °C on 2% malt extract agar. The fungus was cultured at 28 °C in shaken (150 rpm) 250 mL conical flasks with 100 mL of a glucose-peptone medium supplemented with 50 µM MnSO_4_ to produce laccase as unique ligninolytic enzyme, as previously described [33]. Inoculum (1.0 mg dry weight per ml of medium) was prepared by homogenising 7-day-old cultures grown in the same medium and conditions. Culture media were sterilised at 121 °C for 20 min.

### 2.3. Determination of Fungal Biomass and Enzyme Activities

The time course for growth was determined by estimating the dry weight of mycelium. Cultures were vacuum filtered through pre-weighted filters (Whatman no. 1); the mycelium washed three times with MilliQ water and kept at 80 °C until constant weight. Extracellular laccase activity was estimated at room temperature (22 °C) using 5 mM DMP as substrate in 100 mM acetate buffer, pH 4.5, and measuring the production of coerulignone (ε_469_ = 27,500 M^−1^ cm^−1^ when referring to the DMP concentration) [34]. For the determination of cell-bound activities (laccase and QR), appropriate amounts of mycelium washed 3 times with MilliQ water were incubated at room temperature in 100 mL conical flasks containing 40 mL of the following substrate solutions: (i) 5 mM ABTS in 50 mM acetate buffer, pH 4.5, for laccase activity; and (ii) 500 µM BQ in 50 mM phosphate buffer, pH 5.0, for QR activity. Samples were taken at 1 min intervals for 5 min. The mycelium was separated from the liquid by filtration and the pH of the samples for QR activity determinations was lowered to 2.0 with 7.4 M phosphoric acid. Laccase activity was evaluated as the increase in the absorbance at 436 nm (ε_436_ = 29,300 M^−1^ cm^−1^) [38]. QR activity was estimated as the decrease in the absorbance at 247 nm (ε_247_ = 21,028 M^−1^ cm^−1^) [38] of ten-fold diluted samples. Enzyme activities were calculated by regression analysis of data. In previous studies, the reduction of BQ to the corresponding hydroquinone (BQH_2_) was analysed by HPLC [33,39,40]. To avoid underestimations of QR activity, BQ was selected since laccase activity on BQH_2_ has been shown to be quite low [38]. For the present study, we decided to use spectrophotometric analysis after observing in preliminary experiments that, working with washed mycelium, the absorbance at 247 nm in the incubations mostly corresponds to the BQ absorption maximum (incubations blanks were carried out in the absence of BQ). It was also found that the same QR activity results were obtained by analysing the disappearance of quinone by this procedure or by HPLC. International units of enzyme activity (μmol min^−1^) were used.

### 2.4. Quinone Redox Cycling Conditions for ·OH Production 

Induction of ·OH production in *P. eryngii* was performed as follows: mycelial pellets from each culture flask were collected by filtration, washed three times with MilliQ water, and divided in three equal parts (wet weight) to be used as replicates of redox cycling experiments. Incubation mixtures were prepared in 100 mL conical flasks containing 30 mL of 20 mM phosphate buffer, pH 5.0, 500 μM DBQ, and 100 μM Fe^3+^-300 µM oxalate. Iron salt (FeCl_3_) solutions were made up fresh immediately before use. Incubations were carried out in the absence and presence of 100 µM Mn^2+^ (QFe and QFeMn incubations, respectively), and blanks only contained DBQ in the buffer. A common component of all the incubation mixtures was 2.8 mM 2-deoxyribose, since it was the probe used to detect ·OH as the production of TBA-reactive substances (TBARS) [41]. Incubations were performed at 28 °C and 150 rpm. Samples were taken periodically, filtered, and their pH was lowered to 2 with 7.4 M phosphoric acid.

### 2.5. Advanced Oxidation of Dyes

Incubation conditions just described for QRC experiments were used for the advanced oxidation of the dyes, except for 2-deoxyribose which was substituted for the dyes. All the incubations were carried out with washed mycelial pellets (50 ± 5 mg, dry weight) from 7-day cultures and 50 μM of the dyes. To evaluate any adsorption of the dyes to the mycelium, incubations were also performed with sterilised pellets (20 min at 121 °C). Incubations were performed in the dark at 28 °C and 150 rpm. Samples were periodically taken during a studied period of 8 h, filtered, and their pH was lowered to 2 with 7.4 M phosphoric acid to inactive any possible enzyme present. Dyes decolourisation and degradation were analysed by spectrophotometric and chromatographic techniques, respectively, as described below. 

### 2.6. Analytical Techniques

TBARS production from 2-deoxyribose was used as an indirect procedure to estimate extracellular ·OH in samples taken from QRC experiments. For the determination of TBARS, 0.25 mL of 2.8% (wt/vol) trichloroacetic acid and 0.25 mL of 1% TBA in 50 mM NaOH were subsequently added to 0.5-mL samples, which were then heated at 100 °C for 10 min and, after cooling, the absorbance was read at 532 nm against appropriate blanks [41].

The levels of DBQ and DBQH_2_ in QRC experiments were determined by HPLC, using standard calibration curves for each compound from 0.5 to 500 μM (limit of detection in both cases was 0.5 μM). Samples were injected into an Agilent system (1200 Infinity Series, Agilent Technologies, Madrid, Spain), equipped with a reverse phase column (Spherisorb S50DS2, Hichrom Ltd., Reading, UK) and a diode array detector. Analyses were performed at 40 °C with a flow rate of 1 mL min^−1^ using 10 mM phosphoric acid-methanol (80/20) as the eluent. The detector operated at 280 nm and 254 nm for the detection of DBQ and DBQH_2_, respectively.

Decolourisation of dyes was followed by analysis of UV-Vis absorbance scanning in a Hitachi model U-2001 spectrophotometer. UV/Visible absorption spectra at pH 2.0 were performed between 800 and 200 nm, and the wavelength of maximum absorption (λ_max_) in the visible region was determined. Values of determined λ_max_ were used to evaluate the time course of decolourisation of each dye and to determine its extent (%). When required, samples were diluted prior spectrophotometric analyses.

Degradation of the dyes was evaluated by HPLC using the equipment and detector mentioned above for DBQ and DBQH_2_ analyses and standard calibration curves for each compound. The type of columns used and the elution conditions for each dye are described below. The detector operated at 280 nm, 254 nm, and the λ_max_ of each dye. With the reverse phase column Nucleosil C18 (25 cm × 0.46 cm, Scharlau, Barcelona, Spain), Acid Yellow 17, Chromotrope 2R, Crocein Orange G, Orange II, Tropaeolin O, Acid Black 24, and Ponceau SS were eluted at 1.0 mL min^−1^ and room temperature using a linear gradient of acetonitrile in water from 0 to 100% in 10 min. With the reverse phase column Poroshell 120 EC-C18 (2.7 µm, 3.0 × 50 mm, Agilent Technologies, Madrid, Spain), the rest of dyes were eluted, except for the phthalocyanine-type dye, whose appropriate conditions for its analysis could not be determined. Elution conditions were the following: (i) Acid Blue 45, Reactive Black 25, and Indigo Carmine were eluted at 0.7 mL min^−1^ and 45 °C, using 10 mM Na_2_HPO_4_, pH 7, (A) and methanol (B) as elution phases. The gradient used, referred to % B, was 0 min 5%, 0.15 min 5%, 0.5 min 30%, 2.3 min 40%, 2.6 min 40%, 3.3 min 95% and 4 min 95%; (ii) Acid Green 25, Reactive Blue 19, Acid Orange 63, and Methyl Blue were eluted at 0.7 mL min^−1^ and 45 °C, using 10 mM Na_2_HPO_4_, pH 7, (A) and acetonitrile (B) as elution phases. The gradient used, referred to % B, was 0 min 5%, 0.15 min 5%, 0.5 min 30%, 2.3 min 40%, 2.6 min 40%, 3.3 min 95% and 4 min 95%; (iii) Acid Red 88, Methyl Orange, and Tartrazine were eluted in isocratic mode at 1.0 mL min^−1^ and 40 °C, using 10 mM acetic acid (A) and acetonitrile (B) as elution phases. The % of phase B for each of them was 40, 25, and 10%, respectively; (iv) New Coccine was eluted at 1.0 mL min^−1^ and 30 °C, with MilliQ water (phase A) and acetonitrile (phase B). The gradient used, referred to % B, was 0 min 0%, 0.5 min 0%, 1 min 20%, 4 min 20%, 4.01 min 100%, and 5 min 100%; (v) Brilliant Green, Bromophenol Blue, and Cresol Red were eluted at 0.7 mL min^−1^ and 45 °C, using 10 mM phosphoric acid, pH 7, (A) and methanol (B) as elution phases. The gradient used, referred to % B, was 0 min 20%, 2 min 20%, 3 min 40%, 7 min 100%, and 8 min 100%.

### 2.7. Statistical Analyses

All the results included in the text and shown in figures and tables are the mean and standard deviations of three replicates (full biological experiments and technical analyses). 

## 3. Results

The objective of the present research was to demonstrate that the induction of extracellular ·OH in *P. eryngii* could increase its degradative capacity on dyes, not only in terms of the oxidation rate at which its enzymes can oxidise them, but also by extending the number of dyes susceptible to degradation. For this purpose, first, the appropriate fungal culture conditions were selected and the activity of the ligninolytic enzyme produced on the dyes was determined spectrophotometrically (colour loss). Then, conditions were established to induce the production of extracellular ·OH, and the advanced oxidation of the dyes was evaluated, not only spectrophotometrically, but also chromatographically (HPLC). 

### 3.1. Determination of Culture Conditions for Both Dyes Decolourisation and Induction of ·OH Production

To select appropriate culture conditions for this study, *P. eryngii* was cultivated in a complex medium containing glucose and peptone, as carbon and nitrogen sources, respectively, and Mn^2+^ [33]. Under these conditions, the only ligninolytic enzyme produced by the fungus was laccase. Figure 1 shows, in addition to extracellular and cell-bound laccase levels during 11 days of growth, time courses of biomass (estimated as mycelium dry weight) and the activity of QR, which was always cell bound. As can be seen, the growth phase of the fungus extended until the ninth day, when the dry weight yielded its maximum value (432.1 ± 86.4 mg). The QR activity was detected on all sampling days, also reaching its maximum value on day 9 (9.8 ± 2.0 total units -TU-). Regarding laccase activity, the highest levels were detected in the extracellular medium, reaching its maximum value after 7 days of culture (0.6 ± 0.07 TU, or 60 mU mL^−1^). The activity of mycelium-associated laccase presented lower and constant levels between 5 and 7 days of culture, with the highest levels being determined after 9 days (0.3 ± 0.13 TU), coinciding with the maximum dry weight of mycelium. On day 11, the levels of both laccase activities decreased to values detected on the initial days. 

### 3.2. Decolourisation of Dyes by Fungal Culture and Laccase

To determine the ability of *P. eryngii* to decolourise the 27 dyes under study, the dyes were added at a concentration of 50 μM to independent flasks on day seven, i.e., the day of maximum extracellular laccase production. The decolourisation percentage was estimated after 3 h of incubation using UV/Vis spectrophotometry. To distinguish between colour loss due to dye degradation or adsorption to the mycelium, incubations with sterilised mycelium at 121 °C for 20 min were used as a control. The results obtained are summarised in Table 1. 

It should be noted that 10 of the 27 dyes were adsorbed to the mycelium of the fungus. The rest of the dyes tested were decolourised by the fungus, to a greater or lesser extent, except for the azo dyes Acid Yellow 17 and Tartrazine. The best results were obtained with Indigo Carmine and Brilliant Green, showing decolourisation percentages of 76 and 85%, respectively. The decolourisation percentages with the rest of the dyes were lower, and the majority with values between 12 and 57%. 

In order to search if laccase was the enzyme responsible for the oxidation of the dyes, an in vitro decolourisation screening was performed using partially purified fungal laccase from *P. eryngii*, containing the two isoenzymes produced by this fungus [38]. The reaction mixtures contained 300 mU mL^−1^ of enzyme, that is, 5 times more than the extracellular activity found in the fungal cultures (60 mU mL^−1^). The decolourisation capacity of the enzyme on the 27 dyes was also determined after 3 h of reaction (Table 1). Except for Acid Yellow 17, New Coccine and Tartrazine (azo), Acid Orange 63 and Ponceau SS (diazo), Neutral Red (heterocycle), and Methyl Blue (triarylmethane), laccase oxidised the rest of dyes. It was even able to oxidise 8 out of the 10 dyes that adsorbed to the mycelium. The two adsorbed dyes that laccase did not oxidise were Acid Orange 63 and Neutral Red. 

### 3.3. Selection of Quinone Redox Cycling Conditions for Induction of ·OH Production

Based on the results shown in previous studies of *P. eryngii* on QRC [33,34], DBQ and the complex Fe^3+^-oxalate were selected to induce ·OH production in the present study for the good results obtained with them, as explained in the Section 1. Incubations were performed in the absence and presence of Mn^2+^ to produce ·OH at two different rates and, thus, be able to better assign the oxidation of the dyes in the following degradative experiments to these radicals. TBARS production from 2-deoxyribose was the indirect method selected for the detection of ·OH. In *P. eryngii* experiments as this method has always been well correlated to the production of components of Fenton’s reagent with other indirect procedures of ·OH detection, such as the hydroxylation of 4-hydroxybenzoic acid to 3,4-dihydroxybenzoic acid, and with the rate of pollutants degradation. In general, the experiments from the studies of *P. eryngii* mentioned above were carried out with 10-day-old, washed mycelium. To find out the optimum culture age to induce ·OH production for the present study, both the whole culture and washed mycelium from 5, 7, 9, and 11 days were incubated at pH 5.0 under the selected QRC conditions, and TBARS production was determined. For whole culture incubations, half of the culture liquid was eliminated and replaced by the solutions of buffer, DBQ, Fe^3+^-oxalate, 2-deoxyribose and, when indicated, Mn^2+^. Incubations were carried out for 2 h and samples taken every 30 min, showing that TBARS were produced on a constant basis the four days tested and under all conditions assayed. In no case was TBARS production observed in the incubation blanks performed in the absence of either DBQ or Fe^3+^-oxalate. As an example, Figure 2A shows the results obtained with the 5-day culture and mycelium.

Regression analysis of the data obtained in incubations without Mn^2+^ rendered TBARS production rates of 16.8 ± 2.4 and 35.0 ± 0.9 mU A_532_ min^−1^ with whole culture and washed mycelium, respectively. In incubations with Mn^2+^, the rates enhanced to 27.1 ± 3.6 and 82.3 ± 1.6 mU A_532_ min^−1^ with whole culture and washed mycelium, respectively. These rates, together with those obtained on the rest of the days tested, are shown in Figure 2B. Regardless of the presence of Mn^2+^, TBARS production rates determined in incubations with washed mycelia were between 1.5 and 3.0 times higher than with whole cultures. Although laccase levels were higher in whole cultures (Figure 1), the lower levels of TBARS detected in these incubations could be explained considering that some components from the culture medium or metabolites produced by the fungus were acting as ·OH scavengers. The presence of Mn^2+^ in the incubations increased TBARS production rates between 1.6 and 3.0 times, obtaining the highest value in incubations carried out with 7-day-old, washed mycelium (91.3 ± 0.8 mU A_532_ min^−1^). Based on these results, the latter was the mycelium age selected for next experiments.

To determine how long the advanced oxidation experiments of the dyes should last, the temporal operability of ·OH production under the QRC conditions selected was studied in long duration experiments. Figure 3 shows the time courses of the concentration of DBQ and DBQH_2_, as well as the sum of both, i.e., DBQ(H_2_), and the production of TBARS for 8 h. Incubations of 7-day-old washed mycelium were performed only with DBQ (control incubation, panel A) and with DBQ and Fe^3+^-oxalate in the absence and presence of Mn^2+^ (panel B and C, respectively). In control incubations, no TBARS were generated due to the lack of the iron complex. In addition, DBQ was rapidly reduced to DBQH_2_ by QR during the first 0.25 h, reaching DBQH_2_/DBQ molar ratios around 6 between 0.5 and 3 h, indicating that oxidation of DBQH_2_ by mycelium-associated laccase was the rate-limiting step of the redox cycle. The concentration of DBQ(H_2_) during this period remained constant around 450 μM. After 3 h, DBQH_2_/DBQ molar ratio and DBQ(H_2_) levels decreased slowly and gradually until the end of the experiment, probably for one or several of the following reasons: (i) variations in laccase and/or QR activities; (ii) participation of other enzymatic or chemical agents in the redox cycle; and (iii) DBQ(H_2_) consumption by the fungus. In this regard, it should be noted that the intracellular metabolism of aromatic compounds, such as DBQH_2_, has been described in WRF involving monooxygenases [42], which hydroxylate aromatic compounds as a previous step of the cleavage of benzene ring by dioxygenases [43]. 

In incubations performed with Fe^3+^-oxalate, without and with Mn^2+^ (panels B–C), reduction of DBQ to DBQH_2_ was also observed, but the DBQH_2_/DBQ molar ratios were much lower than in the control incubations, mainly due to the fast disappearance of DBQ(H_2_) and the production of ·OH, which was evidenced by TBARS generation. As can be seen, these two parameters were quite well correlated, indicating that ·OH were responsible for the DBQ(H_2_) degradation. To better understand the last statement, let us analyse these results in more detail. Initial TBARS production rates (calculated from the values obtained in the range 0–1 h) were 43.6 ± 2.5 and 95.1 ± 10.2 mU A_532_ min^−1^ in incubations without and with Mn^2+^, respectively (similar values to those shown in Figure 2B with 7-day-old mycelia). These different rates of TBARS production were in good agreement with the following parameters regarding DBQ(H_2_) in the corresponding incubations (without and with Mn^2+^, respectively): degradation rates during the first hour of 185.8 and 334.1 μM h^−1^, degradation percentages after 2 h of 60 and 92%, and complete degradation at 4 and 8 h, justifying why TBARS production ceased after these times. Based on these results, the maximum extent of advanced oxidation of dyes experiments was fixed in 8 h. 

### 3.4. ABOP of Dyes, Mediated by P. eryngii 

In experiments of decolourisation carried out with *P. eryngii* cultures (Table 1), it was observed that ten of the dyes tested were adsorbed to the mycelium of sterilised cultures. Sterilised controls for advanced oxidation experiments of dyes were carried out with washed mycelium, finding in this case that Tropaeolin O was not adsorbed. To avoid the adsorption of the remaining nine dyes, a surfactant was added to the QRC incubation mixtures that would increase the solubility of the compounds. The chosen surfactant was Tween 20. Prior to its use in the incubations, it was necessary to determine which concentration was the most suitable, considering not only the one that prevented adsorption of the dyes to the mycelium, but also the one that sequestered the least ·OH. Using different concentrations of Tween 20 in the range 0.05 and 1.00% (*v*/*v*), it was observed that the rate of TBARS production in incubations of the fungus with DBQ and Fe^3+^-oxalate, with and without Mn^2+^, decreased from 25 to 85%, respectively. Thus, it was decided to use the lowest concentration assayed (0.05%). This concentration prevented the adsorption to the mycelium of four more dyes, i.e., Acid Red 88, Acid Black 24, Acid Orange 63, and Cresol Red. 

Advanced oxidation of the dyes by *P. eryngii* was first analysed spectrophotometrically. Decolourisation time courses were followed at the wavelength of maximum absorption in the visible spectrum of each dye. UV/Vis spectra of samples were carried out periodically for 8 h. Figure 4 shows the results obtained with the azo dye Orange II, including pictures of incubation samples taken at 0 and 8 h to observe the loss of colour. These results are representative of those obtained with most dyes. As can be seen in panel A, decolourisation was mainly observed when the fungus was incubated under QRC conditions, both in the absence (QFe) and presence of Mn^2+^ (QFeMn). In the control experiment, carried out only with buffered solutions of the dye and the fungus, a slight decrease in the A_485_ was produced, reaching 8% at the end of the incubation period. In good agreement with the results shown in Figure 3, Orange II decolourisation was faster in the presence of Mn^2+^, coinciding with the faster production of TBAR and DBQ(H_2_) degradation. Calculating the rate of these three parameters during the first hour of incubations, it was always 1.5 to 3 times faster in QFeMn than in QFe incubations. The stabilisation of the colour loss of the dye coincided in time with the consumption of DBQ(H_2_) and the consequent stop in the production of TBARS, approximately at 4 and 8 h in the incubations carried out in the presence and absence of Mn^2+^, respectively. The UV/Vis spectra of the samples taken from QFeMn incubations (panel B) revealed at zero-time two absorption maxima, one in the visible spectrum corresponding to the dye and a second one in the UV spectrum, around 280 nm, mainly due to DBQ (separate spectra of these compounds confirmed the above assertion). It can be observed that throughout the incubations not only the maximum of the dye disappeared but also that of DBQ, according to the results shown in Figure 3C. The loss of the initial orange colour of QFeMn incubations is shown in panel C. 

The results obtained with the rest of the dyes that did not adsorb to the mycelium are shown in Appendix A, using the same presentation format, with the exception that only the spectra of the visible region are shown, so that the changes produced in the dyes can be better observed on a shorter absorbance scale. In this context, it is worth mentioning that no significant differences in the time courses of disappearance of the DBQ absorption maximum were observed, indicating that in all the incubations of the dyes, carried out under QRC conditions, it was mostly degraded, according to the results shown in Figure 3B,C. Among the results shown in Appendix A are those of the control incubations carried out with buffered solutions of the dyes and washed mycelia. As mentioned in the case of Orange II, the percentages of decolourisation of the rest of dyes after 8 h in control incubations were around 8%, except for 5 dyes that exceeded 20%: Acid Black 24 (25%), Acid Orange 63 (42%), Iron(III)phthalocyanine (56%), Brilliant Green (91%), and Indigo Carmine (97%). Nevertheless, the effect of inducing ·OH production in the incubations with these five dyes could be well observed, except in the case of Brilliant Green, whose decolourisation in the control incubations was as rapid as in the incubations carried out under QRC condition. The low decolourisation percentages observed in the control incubations of most of the dyes, when compared to those determined in the cultures (Table 1), which generally reached values above 20% in three hours, indicated that some decolourisation agents were lost when the mycelium was washed. Anyway, working with washed mycelium facilitated the study of the effect of inducing in the fungus ·OH production on the decolourisation of the dyes, since the differences observed between the control and QRC incubations were greater. To ascribe the oxidation of the dyes in the control incubations solely to the laccase activity that remains associated with the mycelium after washing it was not possible because among the 5 dyes mentioned, some of them were oxidised by the enzyme at relatively high rates (e.g., Brilliant Green), while others, such as Indigo Carmine, were oxidised at very low rates (Table 1). The results shown in Appendix A illustrate quite well the loss of the absorption maximum of visible light of all the dyes by the action of the radicals produced in the QFeMn incubations (panel B) and, consequently, the loss of their characteristic colours (panel C).

To complement the results of Appendix A and to better analyse them, Table 2 shows the percentages of decolourisation determined in samples taken after 1, 3, and 8 h from QFe and QFeMn incubations. Due to the rate at which ·OH were produced and DBQ(H_2_) consumed over time (Figure 3, panels B–C), the highest decolourisation rates occurred during the first hour of incubation. Considering the results obtained with all the dyes, the average values of the decolourisation percentages after one hour in control, QFe, and QFeMn incubations were 7, 37, and 51%, respectively (5.2 and 7.3 times higher in QFe and QFeMn incubations, respectively, respect to control). To calculate how much of these percentages were due to ·OH radicals, the decolourisation percentages observed at the same times in the control incubations (Appendix A) were subtracted from each of the values shown in Table 2. Thus, the average values of decolourisation due to ·OH were estimated to be 30 and 47% in QFe and QFeMn incubations, respectively. That is, the presence of Mn^2+^ increased the rate of decolourisation 1.6 times, in agreement with the effect of this cation on ·OH production (Figure 3, panels B–C). The effect of Mn^2+^ is clearly observed in Appendix A, except in incubations with Acid Blue 45, Acid Red 88, and Brilliant Green, where decolourisation occurred at similar rates in QFe and QFeMn incubations. After 3 h, the average decolourisation percentages, corrected for the values of the control incubations, were 64 and 74% in QFe and QFeMn incubations, respectively. Although the decolourisation was enhanced in both incubations, the increase with respect to the percentages obtained at one hour was slightly lower in QFeMn incubations (27 vs. 34% without Mn^2+^). This fact can be explained by considering that in QFeMn incubations, the maximum decolourisation was around to be reached (Appendix A) due to the exhaustion of OH production, as shown in Figure 3C. Finally, after 8 h of incubation the average values of the decolourisation percentages caused by ·OH were similar in QFe and QFeMn incubations (71 and 73%, respectively), indicating that the decolourisation caused by ·OH could reach the end and proving that at 3 h in QFeMn incubations it had already ended.

The degradation of the dyes in samples from QFeMn incubations was also evaluated by HPLC, together with DBQ(H_2_) levels, which are indicative of ·OH production, to see if the two parameters were well correlated. Figure 5 depicts the result obtained with the anthraquinone dyes Acid Green 25 and Acid Blue 45, since they illustrate the two degradation profiles observed. In both cases, complete DBQ(H_2_) degradation occurred during the first 2–3 h, showing a very-high and similar rates during the first hour, i.e., 439 ± 37 and 466 ± 48 μM h^−1^, respectively. Similar degradation rates of DBQ(H_2_) were calculated with all the dyes tested. In the incubation with Acid Green 25, the dye was degraded in parallel with DBQ(H_2_), reaching a degradation rate of 36 ± 3 μM h^−1^. For an adequate comparison with DBQ(H_2_) degradation rate, it should be considered that the initial concentration of the dye was 10 times lower than that of DBQ(H_2_). After 3 h of incubation, DBQ(H_2_) was completely degraded, leaving 6 ± 2% of undegraded dye, which gradually decreased to 2 ± 2% after 8 h. The results obtained with Acid Green 25 were characteristic of most of the dyes tested. With respect to Acid Blue 45, it showed a degradation rate during the first hour of 17.6 ± 0,4 μM h^−1^ (twice as low as that observed with Acid Green 25), despite the high degradation rate of DBQ(H_2_). After the complete degradation of the later (2 h), the percentage remaining of the dye was 56 ± 6%, which gradually decreased to 50 ± 7% at 8 h. This slow rate of degradation of the two dyes when the production of ·OH ceases may be due to the action of mycelium-associated laccase which, as shown in Table 1, was active on both dyes. As for the slow degradation rate of Acid Blue 45 from the beginning of the incubation, one possible explanation could be that this dye has more than one site for ·OH attack, or that the number of intermediate degradation fragments derived from ·OH action is higher than with other dyes, thus increasing the competition of these radicals for the attack of the parent compound molecules. Similar results to those shown with Acid Blue 45 were also observed in the case of New Coccine and, with higher initial rates and percentages of degradation, with Acid Black 24 and Acid Orange 63.

The chromatograms of Acid Orange 63 and the rest of dyes studied are presented in Appendix A. The high efficiency of the process should be highlighted as out of the 21 dyes analysed, of different structures (Appendix A), 11 of them were completely degraded after 2 h of incubation, with another 8 dyes reaching values between 89 and 99%, and only 2 of the dyes were partially degraded (around 50%). The exact degradation percentages and times at which they were reached have been added to Table 2. As can be seen, comparing these percentages with those of decolourisation achieved in the same samples (8 h in QFeMn incubations), the spectrophotometric analysis generally showed lower values, since in the incubations there was always some colour remaining. This could be caused by undegraded coloured degradation intermediates or by coloured contaminating compounds of the commercial dyes (note the low dye content of some of them in the Section 2). The influence of possible metabolites produced by the fungus on the colour of the incubations cannot be ruled out either.

## 4. Discussion

### 4.1. From the Use of WRF and Their Enzymes for Pollutant Degradation to ABOP Mediated by These Fungi

A paradigmatic event on the ability of microorganisms to degrade recalcitrant xenobiotics occurred in 1985 when it was shown that a single species of WRF, *P. chrysosporium*, was able to mineralize several persistent environmental pollutants of very different structure, involving its extracellular ligninolytic system [44]. The interest that this finding triggered in the scientific community is currently reflected in the large number of review papers of a general nature in the field of pollutants degradation by WRF [2,16,45] or referring to specific groups of pollutants [46,47,48,49], fungal species [50,51], and ligninolytic enzymes [52,53]. This research ran parallel to the development of the knowledge that was being generated on the mechanisms of degradation of lignocellulosic materials by these fungi. Thus, the demonstration of the involvement of low molecular weight ligninolytic agents to degrade wood opened the door to the use of these agents for decontamination. Most research in this field focused on the Mn^3+^ generated by some ligninolytic peroxidases [54] and, above all, on fungal metabolites and lignin depolymerization products which act as mediators of the ligninolytic enzymes, mainly laccase [53]. In relation to ·OH, the two mechanisms described for their production with the ligninolytic enzymes (LiP and laccase) were more complex because they require a higher number of reagents and the enzymes acted as an indirect source of ·OH production through the generation of Fe^3+^ and O_2_ reductants from the oxidation of adequate fungal metabolites or lignin-derived depolymerization products [27,28]. In these ·OH production mechanisms, contrary to what ideally occurs with the enzyme–mediator systems, the electron donors (enzyme substrates) are depleted, so that, for their use in the degradation of pollutants, it would imply their continuous addition to the reactions to complete the degradative process. This is probably one of the reasons why, to our knowledge, there are no studies on the use of these enzymatic ·OH production systems to degrade pollutant compounds. However, in the case of the mechanism involving laccase and hydroquinones, it was demonstrated that this constraint was no longer a limitation when operating in vivo, since WRF are characterised by the production of a set of intracellular dehydrogenases [55], including those reducing quinones, whose products were secreted into the extracellular medium leading to the establishment of redox cycles, such as QRC [33]. 

Thanks to the description of this QRC mechanism, several studies, including the present one, have been able to evaluate the effect of inducing the production of ·OH on the degradative capacity of various WRF. The fact that this natural process of ·OH production is inducible, together with the possibility of being able to exert some control over the levels of the ·OH radicals produced, makes it, when used as a strategy to degrade pollutant compounds, an ABOP like other AOP based on the Fenton reaction. Using the same QRC conditions to those used in the present work, several studies on ABOP mediated by *T. versicolor* have been described. Aranda et al. [40] reported the degradation of aromatic pollutants as recalcitrant as benzene, toluene, ethylbenzene, and xylene isomers (BTEX), only when *T. versicolor* was incubated under QRC conditions. It is worth mentioning that BTEX degradation by WRF had been only demonstrated with *P. chrysosporium* with the non-involvement of the ligninolytic enzymes [56]. In a parallel study with *T. versicolor*, Marco-Urrea et al. [39] demonstrated the advance oxidation of four chlorinated aliphatic hydrocarbons (trichloroethylene, perchloroethylene, 1,2,4- and 1,3,5-trichlorobenzene), showing the almost complete dechlorination of the compounds and the mineralization of trichloroethylene. The same ABOP mediated by *T. versicolor* was used for the first time to degrade the pharmaceuticals clofibric acid, carbamazepine, atenolol, and propranolol [57]. It is worth noting that the identified hydroxylated intermediates in this study were observed to disappear completely as the incubations progressed. The degradation of a larger number of pharmaceuticals with this *T. versicolor*-mediated ABOP has been validated in subsequent studies using synthetic and real wastewater effluents, with the fungus immobilised in a rotating biological contactor [58] and hospital wastewater [59]. These and other ABOP studies have been recently reviewed, together with an interesting compilation of bioremediation strategies combining WRF with different AOPs based on Fenton reactions (conventional and heterogeneous Fenton-like, photo- and electro-Fenton), which allows for comparisons of the different strategies [36].

### 4.2. ABOP of Dyes Mediated by P. eryngii 

In the present study, we have extended the WRF-mediated ABOP based on QRC to dyes degradation. To date, only the dye phenol red had been shown to be degraded in such a biological process [34]. To evaluate if QRC increases the capability of *P. eryngii* to degrade the selected dyes, their decolourisation was first tested in cultures in which the fungus only produced laccase, and the activity of this enzyme on the dyes was determined (Table 1). As shown in numerous studies on textile dyes’ decolourisation by WRF [49], *P. eryngii* decolourised in three hours most of the dyes that were not adsorbed to the mycelium (13 from 17 non-adsorbed dyes) by more than 10% (37% average). Comparing the percentages of decolourisation obtained in laccase reactions, it seemed clear that laccase was involved in the oxidation of the dyes in the cultures when its activity in the in vitro reactions was relatively high, such as Acid Green 25 and Reactive Blue 19 (anthraquinone), Iron(III)phthalocyanine, and Brilliant Green (triarylmethane). However, with some compounds decolourised in the cultures, laccase activity was very low, mainly with the azo dyes. This suggested that there must be other mechanisms involved in the decolourisation process. One possible explanation is the presence in cultures of fungal metabolites acting as laccase mediators, since *Pleurotus* species, like most WRF, are characterised by the production of a wide variety of phenolic compounds, e.g., 4-hydroxybenzoic acid, which could play this role [60]. 

Keeping in mind the aim of this study, the conditions used to evaluate the effect of inducing in *P. eryngii* the production of ·OH on the decolourisation and degradation of dyes were carefully selected. These conditions should ensure, firstly, that the production of ·OH is as high as possible and, secondly, that any factors interfering with the oxidation of the dyes by these radicals are minimised as much as possible. Thus, the growth of the fungus was carried out in the presence of Mn^2+^ to repress the synthesis of VP [34]. It has been previously shown that when VP is present in cultures, ·OH levels derived from QRC decrease because oxidation of hydroquinones by VP consumes part of the H_2_O_2_ required for ·OH generation [33]. In addition, VP also has the ability of decolourising several dyes [61], so its presence would make it difficult to attribute the decolourisation of the dyes to ·OH radicals. For the determination of QRC conditions, the optimal age of mycelium was determined using the whole culture and washed mycelium, reaching the highest levels of TBARS with 7-day-old washed mycelium in QFeMn incubations (Figure 2). In previous studies on QRC in *P. eryngii* carried out under the same conditions (QFeMn incubations), but with 10-day-old washed mycelium, the specific TBARS production rate was 0.45 mU A_532_ min^−1^ mg^−1^ [34]. In the present study with 7-day-old washed mycelium, this rate increased 4 times to 1.83 mU A_532_ min^−1^ mg^−1^. Due to the high reactivity and low selectivity of ·OH radicals it is likely, as mentioned above, that the lower levels of TBARS observed with whole cultures were due to the reaction of these radicals with other compound and metabolites present in cultures. This possibility was also considered for the selection of QRC conditions because oxidation of such putative compounds and metabolites by ·OH could produce secondary radicals which could oxidise the dyes, leading to underestimations of the direct action of ·OH on the dyes. The use of washed mycelium to evaluate the involvement of ·OH in the oxidation of dyes also presented the advantage of reducing laccase levels in cultures to those remaining bound to the mycelium. As in the case of VP, the presence of higher levels of laccase, which was able to oxidise many of the dyes tested (Table 1), would have made it more difficult to attribute the degradation of the dyes to ·OH radicals. Although the decreased laccase levels in washed mycelium would reduce the rate of the quinone redox cycle, chemical oxidation of DBQH_2_ by Fe^3+^-oxalate prevented this from occurring. 

The experiments carried out to evaluate the advanced oxidation of dyes by *P. eryngii* have clearly demonstrated, on the one hand, that QRC enabled the degradation of dyes that were not decolourised by the fungus in the cultures, such as the azo dyes Acid Yellow 17 and Tartrazine, and were not substrates of laccase, including, in addition to these two azo dyes, New Coccine, the diazo dyes Acid Orange 63 and Ponceau SS, the heterocycle Neutral Red, and the triarylmethane Methyl Blue. On the other hand, QRC greatly increased the degradation rate of most of them. Discarding the dyes that were adsorbed to the mycelium, this effect was observed with the remaining 22 dyes, except for the triarylmethane Brilliant Green because of its efficient degradation by the fungus without inducing ·OH production. In addition to these initial observations, the following findings should be highlighted from the analysis of the results. First, in agreement with previous studies, the use of TBARS production from 2-deoxyribose to estimate the time courses of ·OH generation, rendered results that were positively correlated to those of dyes decolourisation. Thus, the higher the rate of TBARS production, the higher the rate of dyes decolorization. For instance, the specific initial TBARS production rate in QFeMn incubations was 1.9 times higher than that found in QFe incubations (1.83 vs. 0.95 mU A_532_ min^−1^ mg^−1^, respectively). Additionally, the average of decolourisation percentages obtained at 1 h was 1.6 times higher in QFeMn than in QFe incubations (47 vs. 30%, respectively). Second, in general the decolorization of the dyes was maintained until TBARS production ceased because of the degradation of DBQ(H_2_) by the hydroxyl radicals themselves. Thus, while in the QFe incubations the decolorization process was generally maintained during the 8 h duration of the experiments, in the QFeMn incubations, decolorization of the dyes was terminated at around 3–4 h. And third, the maximum average percentages of decolourisation achieved with all the dyes in QFeMn incubations were around 83%, which corresponded to an average degradation rate of around 93% when the levels of the dyes were determined chromatographically. At this point it, should be noted that 10 of the 20 dyes analysed by HPLC (Brilliant Green is not considered) reached 100% degradation in 2–4 h for most of them. Although the conditions used to degrade dyes with WRF in numerous studies are so different that meaningful comparisons cannot be made, the degradation efficiency of dyes with the ABOP used in the present study is generally much higher [49]. Similarly, if the present results are compared with those obtained with AOPs based on Fenton reactions, degradation percentages between 90 and 100% are a common feature [62].

### 4.3. Perspectives and Future Research 

Among the different degradative agents that WRF produce, ABOPs based on QRC promote the generation and the action of the most powerful of them, i.e., ·OH. The results obtained to date using this degradation process, including those of the present work, have shown that the number and structural diversity of compounds that can be degraded is large (see Appendix A as an example). Due to the high reactivity and low selectivity of ·OH, it is quite likely that any contaminant whose degradation has been previously demonstrated in AOPs, mainly those based on the Fenton reaction, will also be degraded by these fungi when QRC is promoted. This strategy, in addition to taking advantage of the oxidising power of ·OH, uses the high degradative capacity of these fungi, so that with the research required to its development it could become one of the most powerful biodegradation tools. 

The results of this study, as well as those described with *T. versicolor* [39,40,57], in which the washed mycelium of the fungi was used, have shown in some cases the complete degradation of the pollutants at the concentration at which they were used. From these results it can be inferred that the washed mycelium of the fungi, generally obtained at the end of the growth phase, contained sufficient reserves to produce the reduction equivalents required for QR activity. In other words, working with washed mycelium for the advanced oxidation of small amounts of pollutants does not seem to be a limitation. In this context, interesting questions that arise and that our current research is trying to answer are as follows: is the fungus damaged by the action of hydroxyl radicals? If not, could the mycelium of the fungus be reused to degrade higher amounts of pollutants? Our current results are showing that the washed mycelium of the fungus can be reused in successive batches of degradation as effective as the first batch, provided that the fungus is supplied with an energy source between batches to maintain its energetic metabolism and prevent its death.

To achieve the objective of the present study, the action of hydroxyl radicals on dyes has been favoured over that of other degradative agents produced by *P. eryngii*, such as the ligninolytic enzymes. It is clear, therefore, that in order to evaluate the full degradative potential of this and other WRF on pollutants, further studies are needed in which the induction of OH radicals is performed under culture conditions in which other agents that could participate in the degradative process are present. In general, the degradation of contaminants by WRF requires a first phase of fungal growth under culture conditions that favour the production of the enzymes involved in the degradation process. In a second degradative phase, these cultures are placed in contact with the media in which the contaminants are found. To increase the degradative capacity of WRFs, future studies could consider the addition of QRC-promoting reagents at the beginning of the degradative phase to convert the biodegradative process into an ABOP, provided that better degradation results are obtained under these conditions, as shown recently by del Alamo et al. [58]. Using this strategy, the limitations of ABOPs should be similar to those described in studies of pollutants degradation by WRF [2], and future research should attempt to address them. 

## 5. Conclusions

This study demonstrates the capability of the WRF *P. eryngii* to decolourise several anthraquinone, azo, diazo, indigo, phthalocyanine, and triarylmethane dyes when cultured under conditions producing laccase (60 mU mL^−1^) as the only ligninolytic enzyme. Ten of the twenty-seven dyes tested were absorbed to the mycelium, including two heterocycle dyes. From the remaining 17 dyes, 13 of them were decolourised between 12 and 85% in 3 h. In vitro reactions carried out with purified laccase (300 mU mL^−1^) show the typical wide substrate specificity of this kind of enzymes, since it oxidised 20 from the 27 dyes assayed (including one heterocycle dye that was adsorbed to the mycelium), 12 of them being decolourised between 13 and 86% in 3 h. Although these results indicated that laccase may have been responsible for dye oxidation in cultures, in some cases (mainly with azo dyes) dye oxidation was higher in the cultures than in the in vitro reactions, which contained 5 times more laccase activity. It was clear, therefore, that other oxidizing agents present in the cultures, such as putative compounds mediating laccase activity, were involved in the oxidation of these dyes.

As a strategy to increase de degradative capability of the fungus, the production of the strongest oxidant that WRF can generate, i.e., hydroxyl radicals, was induced through a quinone redox cycling process. In order to demonstrate that the ·OH radicals were the main agents oxidating the dyes, conditions for growing the fungus and inducing ·OH production, estimated as TBARS generation, have been optimised (i) to obtain the highest levels of TBARS, (ii) to limit the action of ·OH on compounds different to the dyes, and (iii) to reduce as much as possible the oxidation of dyes by other degradative agents, including the ligninolytic enzymes produced by this fungus. Based on the results described in preceding studies and those obtained in the present investigation, optimal conditions to study advanced oxidation of the dyes were established as follows: incubations of 7-day-old washed mycelium in phosphate buffer, pH 5, with DBQ and Fe^3+^-oxalate, in the absence and presence of Mn^2+^ (QFe and QFeMn incubations, respectively), leading to different ·OH production rates around twice higher with Mn^2+^. The duration of OH production under these conditions depended on the concentration of DBQ-DBQH_2_ couple, which was continuously being reduced due to the attack of the ·OH radicals themselves. Thus, TBARS production lasted 8 and around 3 h in QFe and QFeMn incubations, respectively. From the results obtained in advance oxidation experiments, the following should be highlighted: (i) all the dyes were decolourised, including those that were not decolourised by the fungus in cultures nor oxidised by laccase in in vitro reactions; (ii) the decolourisation rate of most dyes greatly increased, being positively correlated to the rate of TBARS production in QFe and QFeMn incubations; and (iii) the best average decolourisation value obtained in QFeMn incubations, analysed spectrophotometrically, was around 83%, which corresponded to an average degradation value of around 93%, when the levels of the dyes were analysed chromatographically (10 of the 20 dyes analysed by HPLC reached 100% degradation in 2–4 h). These results allow for us to affirm that the induction of extracellular ·OH production in *P. eryngii* is an efficient strategy that considerably increases its degradative capacity and, therefore, they provide useful information for the development of ABOPs mediated by WRP, based on QRC. 

## Figures and Tables

**Figure 1 jof-10-00052-f001:**
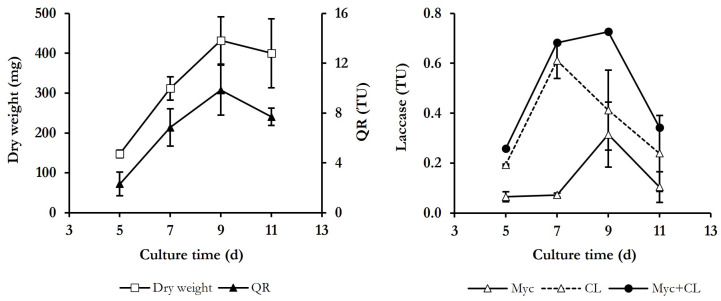
Time course of dry weight, quinone reductase (QR), and laccase, both bound to the mycelium (Myc) and in the culture liquid (CL), in cultures of *P. eryngii*. Total units (TU) of enzyme activities are shown.

**Figure 2 jof-10-00052-f002:**
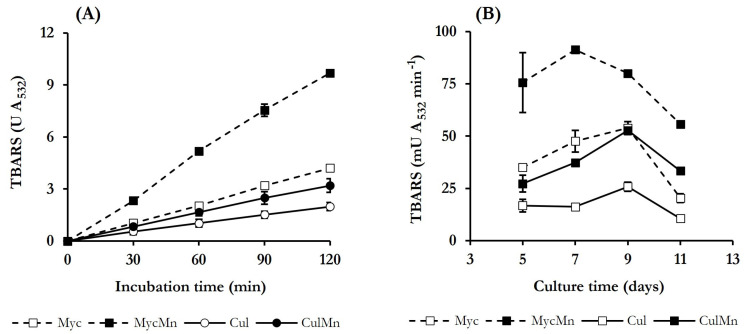
Production of TBARS by *P. eryngii* incubated under quinone redox cycling (QRC) conditions, using whole cultures (Cul) or washed mycelia (Myc): (**A**) Time courses of TBARS production in 5-day-old cultures; (**B**) TBARS production rates as a function of culture age. Incubations were carried out at pH 5.0 (20 mM phosphate buffer) and contained 500 μM 2,6-dimethoxy-1,4-benzoquinone (DBQ), 100 μM Fe^3+^-300 μM oxalate and, when indicated, 100 μM Mn^2+^. Incubation blanks lacked either DBQ or Fe^3+^-oxalate complex.

**Figure 3 jof-10-00052-f003:**
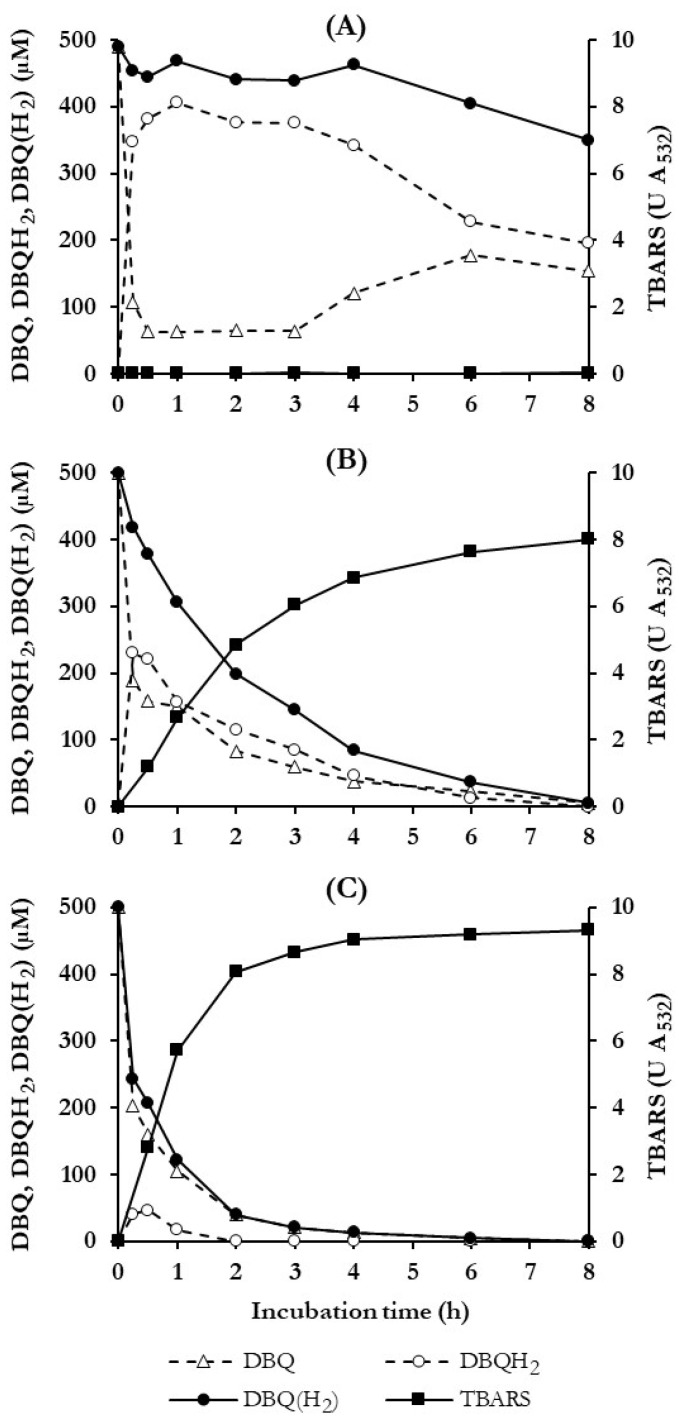
Time course of DBQ, DBQH_2_, the sum of both, i.e., DBQ(H_2_) and TBARS production, during DBQ redox cycling in *P. eryngii*. Incubations contained 30 mL of 20 mM phosphate buffer, pH 5.0, 7-day-old, washed mycelium (50 ± 5 mg, dry weight), 2,8 mM deoxyribose and: (**A**) 500 μM DBQ; (**B**) 500 μM DBQ and 100 μM Fe^3+^-300 μM oxalate; (**C**) 500 μM DBQ, 100 μM Fe^3+^-300 μM oxalate and 100 μM Mn^2+^.

**Figure 4 jof-10-00052-f004:**
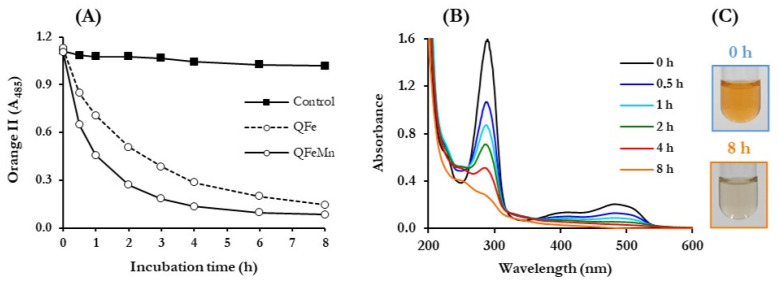
Advanced oxidation of the azo dye Orange II by *P. eryngii*. (**A**) Time course of decolourisation; (**B**) changes in UV/Vis spectra of diluted samples from incubations with Mn^2+^ along the incubation time; (**C**) pictures of QFeMn incubation samples taken at 0 and 8 h. Incubations contained 30 mL of 20 mM phosphate buffer, pH 5.0, 7-day-old, washed mycelium (50 ± 5 mg, dry weight), 50 μM Orange II, 500 μM DBQ, 100 μM Fe^3+^-300 μM oxalate (QFe incubations) and 100 μM Mn^2+^ (QFeMn incubations). Control incubations lacked DBQ, iron complex and Mn^2+^.

**Figure 5 jof-10-00052-f005:**
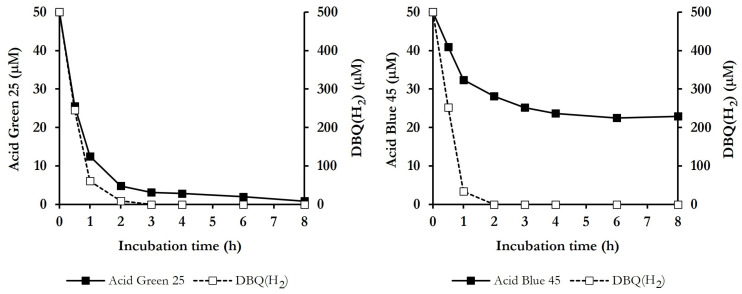
Advanced oxidation of the anthraquinone dyes Acid Green 25 and Acid Blue 45 by *P. eryngii*, showing the time courses of the dyes and DBQ(H_2_) in QFeMn incubations.

**Table 1 jof-10-00052-t001:** Dyes decolourisation in *P. eryngii* cultures and in vitro laccase reactions. Laccase levels in 7-day-old cultures and reactions were 60 and 300 mU mL^−1^, respectively, and the concentration of dyes was 50 μM in both cases. The percentages shown were obtained after 3 h of treatment.

		Decolourisation (%)	
Structure	Dye	Culture	Laccase
Anthraquinone	Acid Black 48	Adsorption	17 ± 1
	Acid Blue 45	14 ± 1	7 ± 0
	Acid Green 25	57 ± 3	86 ± 2
	Reactive Blue 19	22 ± 2	24 ± 1
Azo	Acid Red 88	Adsorption	7 ± 0
	Acid Yellow 17	0 ± 0	0 ± 0
	Chromotrope 2R	29 ± 2	6 ± 1
	Crocein Orange G	19 ± 2	2 ± 0
	Methyl Orange	22 ± 3	4 ± 0
	New Coccine	41 ± 3	0 ± 0
	Orange II	38 ± 4	3 ± 1
	Tartrazine	0 ± 0	0 ± 0
	Tropaeolin O	Adsorption	2 ± 0
Diazo	Acid Black 24	Adsorption	53 ± 2
	Acid Blue 113	Adsorption	73 ± 2
	Acid Orange 63	Adsorption	0 ± 0
	Ponceau SS	12 ± 1	0 ± 0
	Reactive Black 5	1 ± 0	15 ± 1
Heterocycle	Azure B	Adsorption	39 ± 2
	Neutral Red	Adsorption	0 ± 0
Indigo	Indigo Carmine	76 ± 4	2 ± 0
Phthalocyanine	Iron(III)phthalocyanine	44 ± 3	67 ± 2
Triarylmethane	Brilliant Green	85 ± 3	61 ± 2
	Bromophenol Blue	3 ± 1	27 ± 1
	Cresol Red	Adsorption	40 ± 2
	Crystal Violet	Adsorption	13 ± 1
	Methyl Blue	17 ± 2	0 ± 0

**Table 2 jof-10-00052-t002:** Advanced oxidation of dyes by *P. eryngii*, showing the percentages of decolourisation (spectrophotometric analyses) and degradation (HPLC analyses) after 1, 3, and 8 h of treatment in QFe and QFeMn incubations.

		Decolourisation (%)						Degradation (%)
Structure	Dye	QFe			QFeMn			QFeMn
		1 h	3 h	8 h	1 h	3 h	8 h	8 h
Anthraquinone	Acid Black 48 ^1^	Mycelial adsorption						
	Acid Blue 45	16 ± 6	20 ± 6	22 ± 2	17 ± 4	23 ± 2	25 ± 5	54 (8 h)
	Acid Green 25	38 ± 3	61 ± 3	68 ± 6	48 ± 5	67 ± 3	65 ± 1	98 (8 h)
	Reactive Blue 19	18 ± 2	30 ± 3	50 ± 7	28 ± 6	42 ± 7	52 ± 4	93 (8 h)
Azo	Acid Red 88 ^1^	46 ± 5	79 ± 3	83 ± 6	45 ± 2	79 ± 4	89 ± 4	100 (4 h)
	Acid Yellow 17	30 ± 2	68 ± 1	90 ± 1	50 ± 2	87 ± 1	91 ± 0	99 (8 h)
	Chromotrope 2R	29 ± 4	67 ± 2	87 ± 2	50 ± 5	81 ± 1	91 ± 1	99 (8 h)
	Crocein Orange G	38 ± 2	67 ± 1	86 ± 1	60 ± 6	82 ± 3	90 ± 1	100 (4 h)
	Methyl Orange	22 ± 4	56 ± 5	81 ± 0	48 ± 3	82 ± 1	87 ± 1	100 (3 h)
	New Coccine	37 ± 4	78 ± 3	94 ± 0	68 ± 6	91 ± 1	95 ± 0	49 (3 h)
	Orange II	37 ± 2	66 ± 5	87 ± 3	59 ± 4	83 ± 7	92 ± 1	99 (8 h)
	Tartrazine	30 ± 4	56 ± 3	82 ± 2	45 ± 3	78 ± 4	90 ± 2	100 (3 h)
	Tropaeolin O	28 ± 2	55 ± 3	78 ± 1	34 ± 1	68 ± 3	85 ± 2	100 (3 h)
Diazo	Acid Black 24 ^1^	9 ± 2	42 ± 2	72 ± 2	15 ± 3	54 ± 3	78 ± 2	95 (8 h)
	Acid Blue 113 ^1^	Mycelial adsorption						
	Acid Orange 63 ^1^	39 ± 5	65 ± 3	84 ± 4	50 ± 3	74 ± 2	89 ± 0	89 (8 h)
	Ponceau SS	37 ± 1	68 ± 3	84 ± 2	56 ± 3	86 ± 3	89 ± 2	100 (4 h)
	Reactive Black 5	34 ± 4	66 ± 5	86 ± 1	53 ± 3	83 ± 4	90 ± 1	100 (2 h)
Heterocycle	Azure B ^1^	Mycelial adsorption						
	Neutral Red ^1^	Mycelial adsorption						
Indigo	Indigo Carmine	74 ± 1	96 ± 0	97 ± 0	100	100 ± 0	100 ± 0	100 (1 h)
Phthalocyanine	Iron(III)phthaloc.	64 ± 4	75 ± 1	82 ± 1	72 ± 2	82 ± 5	83 ± 4	nd (8 h)
Triarylmehane	Brilliant Green	54 ± 2	78 ± 4	94 ± 3	59 ± 5	80 ± 4	93 ± 8	100 (6 h)
	Bromophenol Blue	34 ± 2	57 ± 1	77 ± 3	52 ± 1	73 ± 1	77 ± 3	91 (8 h)
	Cresol Red ^1^	54 ± 1	87 ± 3	93 ± 1	74 ± 6	91 ± 3	93 ± 1	100 (8 h)
	Crystal Violet ^1^	Mycelial adsorption						
	Methyl Blue	35 ± 2	54 ± 1	71 ± 3	40 ± 1	62 ± 4	74 ± 3	89 (8 h)

^1^ Incubations carried out in the presence of 0.05% Tween 20.

## Data Availability

Data are contained within the article and Appendix A.

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
