# Peer review of "Induction of Extracellular Hydroxyl Radicals Production in the White-Rot Fungus *Pleurotus eryngii* for Dyes Degradation: An Advanced Bio-oxidation Process"

_jof, 2024, doi:10.3390/jof10010052_

Round 1

Reviewer 1 Report

Comments and Suggestions for Authors

At present, environmental organic pollutants represent a significant challenge for human society. Advanced oxidation processes (AOPs) and White rot fungi (WRF) are two categories of organic pollution degradation strategies that have garnered widespread attention. A promising system, which integrates the characteristics and advantages of both AOPs and WRF bioremediation, is termed advanced bio-oxidation processes (ABOPs). Mediated by the quinone redox cycling of WRF, ABOPs have demonstrated considerable potential for the remediation of environmental organic pollutants, showcasing high efficiency, low toxicity, and good controllability. This paper focuses on ABOPs, specifically examining the degradation of various types of dyes under ABOP conditions. The results reveal that, in an ABOP system supplemented with manganese, efficient degradation and decolorization of the dyes were successfully achieved. This research demonstrates the application potential of ABOPs, offering valuable new insights for the efficient remediation of environmental organic pollutants. The whole manuscript is well-structured with clear logic and appropriate discussion. Overall, I recommend its publication, but further revisions and improvements are essential before its formal acceptance. Some suggestions are listed below for consideration:

1.Line 36-40, This section requires further improvement, particularly with additional background information on white-rot fungi (WRF). Additionally, some sentences are unclear in their current form. Below is my attempt at revising these for clarity, provided for your reference (The recommended citations are also listed).

“……attracting the most attention among researchers and technologists. White rot fungi (WRF), a famous group of degraders in nature, possess the capability to degrade both lignin and cellulose biopolymers within lignocellulose biomass. Concurrently, they exhibit unique oxidative and extracellular ligninolytic systems characterized by low substrate specificity, enabling them to transform or degrade diverse environmental contaminants [Science of The Total Environment, 778(2021), 146132; Journal of cleaner production, 354(2022), 131681]. The heightened interest in AOPs and WRF stems from two key attributes desired in degradative agents: high reactivity and low selectivity. These qualities are essential for attacking persistent contaminants and dealing with a broad spectrum and variety of such substances. Both the oxidants produced in AOPs and the enzymes and oxidants generated by WRF exhibit these characteristics, making them highly effective in the remediation of recalcitrant contaminants.”

2.Line 84-85 “To date, five potential mechanisms of extracellular ·OH production in WRF, one chemical [28] and four enzymatic [29-32], have been described”. A more detailed description of these mechanisms, elucidating their specific processes and impacts, rather than merely listing the relevant references, would be beneficial for comprehensive understanding.

3.Line 179, For the determination of quinone reductase (QR) activity, it is encouraged to reference pertinent literature to support the methodology employed in this study.

4.Line 184-201, According to the description in the manuscript, the advanced oxidation of dyes were conducted in phosphate buffer. Compared to culture media containing carbon and nitrogen sources, the production of biomass and enzyme activity in WRF are comparatively weaker in phosphate buffer. Why did the authors not choose a nutrient-rich culture medium that could sustain continuous fungal growth as the reaction system? Could the use of phosphate buffer potentially be a limiting factor in the ABOP reaction? It is recommended that the authors address this in the discussion section of the paper

5.Line 215-220, More details on the HPLC experimental conditions should be provided. Specify the standards used, concentrations chosen, separation conditions, limits of detection of the methods used, columns and equipment used.

6.Line 335-340, In section 3.1, the authors utilized a medium containing glucose and peptone as carbon and nitrogen sources, which exhibited quinone reductase (QR) and laccase activities. However, it raises the question: why is there a need to use washed mycelium here?

7.Line 403, In Figure 3, should the solid circle symbol be designated as DBQ(H2)?

8.In Figure 3, a gradual declining trend is observed for DBQH2. Within the QRC cycle, the oxidation of hydroquinone occurs concurrently with the reduction and formation of quinones, theoretically manifesting in an overall dynamic equilibrium. Although potential reasons for the decrease in DBQH2 are discussed in the manuscript, these explanations lack direct evidence. Could the authors provide more robust experimental evidence to substantiate these interpretations? Alternatively, the author may consider a more comprehensive and in-depth analysis of the factors contributing to the decline of DBQH2 in the Discussion section.

9.Section 4 Discussion, It is recommended to include a discussion on the culture condition used in the ABOP reaction. Specifically, the advantages and limitations of using a nutrient-rich culture medium VS using washed mycelium should be appropriately discussed.

10.It is advisable to include a conclusion section that succinctly summarizes the entire paper, highlighting the significance and value of this study. It is recommended to conduct a brief discussion on the limitations of ABOPs and to provide a prospective outlook on potential future research directions.

11.The overall language and grammar of the article have room for further refinement. Certain sentences exhibit a lack of fluency, necessitating additional improvements in readability.

Comments on the Quality of English Language

The overall language and grammar of the article have room for further refinement. Certain sentences exhibit a lack of fluency, necessitating additional improvements in readability.

Author Response

Comment 1. Line 36-40, This section requires further improvement, particularly with additional background information on white-rot fungi (WRF). Additionally, some sentences are unclear in their current form. Below is my attempt at revising these for clarity, provided for your reference (The recommended citations are also listed).

“……attracting the most attention among researchers and technologists. White rot fungi (WRF), a famous group of degraders in nature, possess the capability to degrade both lignin and cellulose biopolymers within lignocellulose biomass. Concurrently, they exhibit unique oxidative and extracellular ligninolytic systems characterized by low substrate specificity, enabling them to transform or degrade diverse environmental contaminants [Science of The Total Environment, 778(2021), 146132; Journal of cleaner production, 354(2022), 131681]. The heightened interest in AOPs and WRF stems from two key attributes desired in degradative agents: high reactivity and low selectivity. These qualities are essential for attacking persistent contaminants and dealing with a broad spectrum and variety of such substances. Both the oxidants produced in AOPs and the enzymes and oxidants generated by WRF exhibit these characteristics, making them highly effective in the remediation of recalcitrant contaminants.” 

Response 1. The comments are gratefully accepted, and appropriate changes of the unclear sentences indicated by the reviewer in the Introduction section have been made (lines 36-45 of the revised manuscript), including the suggested references. 

Comment 2. Line 84-85 “To date, five potential mechanisms of extracellular ·OH production in WRF, one chemical [28] and four enzymatic [29-32], have been described”. A more detailed description of these mechanisms, elucidating their specific processes and impacts, rather than merely listing the relevant references, would be beneficial for comprehensive understanding.

Response 2. A more detailed description of ·OH production mechanisms has been included in lines 95-122, including that involving laccase and hydroquinones. Consequently, the extracellular reactions of the quinone redox cycling process that had been described in the following paragraph have been removed to avoid repetitions.

Comment 3. Line 179, For the determination of quinone reductase (QR) activity, it is encouraged to reference pertinent literature to support the methodology employed in this study. 

Response 3. The method used for the determination of QR activity was developed in our laboratory for the present study. Working with mycelium extracts or purified enzymes, QR activity is usually determined spectrophotometrically as the oxidation of the electron donor, i.e. NAD(P)H. This method cannot be used working with whole cells. For this reason, in previous studies in which I was the supervisor (references 33, 39 and 40), intracellular QR activity was determined by HPLC as the consumption of quinone or the production of hydroquinone in the incubation liquid. This is possible because the intracellular product of QR reduction is secreted into the extracellular medium. To simplify the analysis, for the present work we found that, when working with washed mycelium, the QR activity can be determined by measuring the decrease in absorbance at the maximum of quinone abortion. This absorbance maximum and the corresponding molar extinction coefficient of BQ was determined in a previous study that we have referenced (38). Changes made to the manuscript to clarify the development of this method can be found in lines 226-237.

Comment 4. Line 184-201, According to the description in the manuscript, the advanced oxidation of dyes were conducted in phosphate buffer. Compared to culture media containing carbon and nitrogen sources, the production of biomass and enzyme activity in WRF are comparatively weaker in phosphate buffer. Why did the authors not choose a nutrient-rich culture medium that could sustain continuous fungal growth as the reaction system? Could the use of phosphate buffer potentially be a limiting factor in the ABOP reaction? It is recommended that the authors address this in the discussion section of the paper.

Response 4. A new paragraph explaining the selection of the culture and QRC conditions for the degradation of dyes, has been including in the discussion section (lines 692-721). As explained, these conditions were the ones we considered most appropriate to achieve the objective of the study. The use of a nutrient-rich medium to sustain the growth of the fungus was not necessary since the induction of ·OH production was done once the fungus had been previously grown. What could be needed was a carbon source to keep the energetic metabolism of the fungus and avoid its depth. However, this was not de case because the fungus had enough nutrient reserves to sustain the production of NAD(P)H required for QR activity and generate ·OH enough to degrade the dyes. Therefore, it seemed that the used of washed mycelium and phosphate buffer was not a limitation. This aspects and suggestions on how to develop ABOPs in future studies are commented in a new section of the manuscript (“4.3 Perspectives and future research”) which have been added (see lines 767-797). The first paragraph of this section is the last paragraph of section 4.2 of the first manuscript.

Comment 5. Line 215-220, More details on the HPLC experimental conditions should be provided. Specify the standards used, concentrations chosen, separation conditions, limits of detection of the methods used, columns and equipment used. 

Response 5. The HPLC equipment, the reversed-phase column and the separation conditions used were already described in section 2.6 of the first manuscript. Therefore, in the revised manuscript we have added the concentrations used to calculate the calibration curves (0.5-500 mM) and the limit of detection, which was at least equal to the lowest concentration used (lines 270-271).

Comment 6. Line 335-340, In section 3.1, the authors utilized a medium containing glucose and peptone as carbon and nitrogen sources, which exhibited quinone reductase (QR) and laccase activities. However, it raises the question: why is there a need to use washed mycelium here?

Response 6. The reasons for using washed mycelium are now exposed in the new paragraph added in section 4.2 (lines 692-721).

Comment 7. Line 403, In Figure 3, should the solid circle symbol be designated as DBQ(H2)? 

Response 7. We thank the reviewer for the detection and communication of this mistake. Legend of Figure 3 has been corrected.

Comment 8. In Figure 3, a gradual declining trend is observed for DBQH2. Within the QRC cycle, the oxidation of hydroquinone occurs concurrently with the reduction and formation of quinones, theoretically manifesting in an overall dynamic equilibrium. Although potential reasons for the decrease in DBQH2 are discussed in the manuscript, these explanations lack direct evidence. Could the authors provide more robust experimental evidence to substantiate these interpretations? Alternatively, the author may consider a more comprehensive and in-depth analysis of the factors contributing to the decline of DBQH2 in the Discussion section. 

Response 8. We do not have experimental evidence to substantiate the possible reason given for the decline observed in control incubations of DBQ(H2) levels after 3 hours of incubation (panel A). Therefore, we have only commented that intracellular degradation is possible by oxygenases previously described in WRF (lines 431-435).

Comment 9. Section 4 Discussion, It is recommended to include a discussion on the culture condition used in the ABOP reaction. Specifically, the advantages and limitations of using a nutrient-rich culture medium VS using washed mycelium should be appropriately discussed.

Response 9. As mentioned above, these aspects are now discussed in sections 4.2 (lines 692-721) and 4.3 (lines 767-797). Besides, it should be noted that the use of a nutrient-rich medium in the ABOP reactions, promoting the primary metabolism and the growth of the fungi could reprime the synthesis of ligninolytic enzymes, as some of them are only produced during secondary metabolism, when a nutrient (usually nitrogen) is limiting the growth of WRF.

Comment 10. It is advisable to include a conclusion section that succinctly summarizes the entire paper, highlighting the significance and value of this study. It is recommended to conduct a brief discussion on the limitations of ABOPs and to provide a prospective outlook on potential future research directions. 

Response 10. A conclusion section has been included (lines 798-840) and how to develop ABOPs without increasing the limitations that WRF presents for their use in bioremediation strategies is suggested (lines 781-797).

Comment 11. The overall language and grammar of the article have room for further refinement. Certain sentences exhibit a lack of fluency, necessitating additional improvements in readability.

Response 11. We have tried to improve the fluency of some sentences and the readability of the text, mainly by shortening sentences that were too long.

Reviewer 2 Report

Comments and Suggestions for Authors

Ms. jof-2771238 deals with the decolourization of several dyes belonging to seven chemical classes (e.g., azo, diazo, anthraquinone and so no) by an advanced biooxidation process relying on the generation of hydroxyl radicals via a P. eryngii-based quinone redox cycling (QRC) system. Decolorization was remarkably enhanced by including bivalent manganese in the reaction mixtures.

The Manuscript is written in a clear and concise form. The experimental setup and the methodological approaches are adequate to support the Authors’ conclusions. Although this is not the first time that a QRC system has been applied to dye decolourization, the work has the merit of having extended the investigation to a wide range of dyes, characterized by very diversified chemical structures and of having included a whole series of controls, such as those aimed at establishing the extent of mycelial adsorption and the ability of some isolated components of the QRC system to oxidize the target dyes. Moreover, as opposed to other studies, the actual dyes bioconversion has also been assessed by reverse-phase HPLC. 

For these reasons, I believe the Ms. can be accepted with minor revisions.

Remark #1: I wonder whether, in the TBARS assay, leading presumably to the production of malondialdehyde (MDA), results might be provided in molar amounts using the molar exctinction coefficient of MDA (please, see Figure 2).. 

Table 5: Change “micelial” to “Mycelial”,

Table 5: Change “Degradatión” to “Degradation”

Table 5: Use the extended names for “Phthalocyanines” and “Triarylmethanes”  

Author Response

Remark #1: I wonder whether, in the TBARS assay, leading presumably to the production of malondialdehyde (MDA), results might be provided in molar amounts using the molar exctinction coefficient of MDA (please, see Figure 2). Molar absorption coefficient of MDA at 535 nm of 1.49 x 105 M-1 cm-1

Response 1. We appreciate the reviewer's suggestion. However, we prefer to express the TBARS production results as the increase in Absorbance at 532 for two reasons: first, because we are not sure that all the fragments produced by hydroxyl radical attack to 2-deoxyribose are malondialdehyde; second, because using the same units as in previous studies allows us to compare the results without having to perform further conversion, as shown now in lines 704-708 of the revised manuscript. 

Table 5: Change “micelial” to “Mycelial”.  Corrected

Table 5: Change “Degradatión” to “Degradation”. Corrected

Table 5: Use the extended names for “Phthalocyanines” and “Triarylmethanes”.  Corrected

Reviewer 3 Report

Comments and Suggestions for Authors

Comments to the author

Author has prepared Induction of extracellular hydroxyl radicals production in the white-rot fungus Pleurotus eryngii for dyes degradation: an advanced bio-oxidation process

The studies were well carried over, the figures and relevant information’s are well structurally organized and arranged. The amount of work quite large and tremendous I have few concerns and comments that need to be clarify/justify before prior to publication.

Author Response

Comment 1. In discussion/introduction section, author should discuss more details about white-rot fungus Pleurotus eryngii related materials, advantages and disadvantages, why author chosen this material as a photo catalyst, how the author has improved from the previous studies to till date, please support with catalytic efficiencies with previously reported materials and discuss your current studies?

Response 1. We apologize for the omission of some basic knowledge about Pleurotus eryngii. The information required to understand the study we have done in the present work was described in detail in the first version of the manuscript. In fact, P. eryngii was chosen precisely because of the deep knowledge we have about the induction in this fungus of extracellular production of hydroxyl radical, via a quinone redox cycling process, which was first described by members of our current research group. Anyway, a new reference has been included in the revised version of the manuscript, showing many characteristics of P. eryngii, including the wide array of enzymes it produces for the degradation of lignocellulosic materials (lines 126-128, reference 32).

In the discussion section it is now indicated that the specific rate of ·OH production (estimated as TBARS generation) has been improved 4 times with respect to previous studies on quinone redox cycling in P. eryngii (lines 704-708).  

Comment 2. Author has to discuss the previously reported similar photocatalytic materials with catalytic efficiencies for 7 type of dyes followed by your proposed catalyst catalytic efficiencies how you improved from the existing studies, the advantage current studies this has to be discussed in more detail. I suggest author has to provide comparison table.

Response 2. P. eryngii is a fungus that cannot be considered a photocatalytic material. Therefore, comparisons in the terms proposed by the reviewer cannot be established.

Comment 3. Author has to perform their regeneration studies through their suitably selecting their desorbing solution and how many cycles that can achieved, I advise them to showcase their results for before and after regeneration.

Response 3. The proposed biological model does not require a regeneration process equivalent to a chemical catalyst. What can be done is to reuse the fungus in successive degradation batches. In fact, this is part of our current research, as mentioned in lines 776-780.

Comment 4. Please author need to be more careful, during the catalytic degradation processes, what are the proposed mechanism, why this catalyst can able to degrade the 7 type of dyes organic molecules, what are the insights of this material.

Response 4. Schemes of quinone redox cycling process leading to the production of extracellular ·OH radicals have already been shown in our previous works (references 33-34). This is now mentioned in the revised version of the manuscript (lines 169-171).

The degradation of dyes, regardless of their structure, as well as many other organic and inorganic pollutants, by ·OH radicals is due to its high reduction potential (2.8 V) and low selectivity. This is the reason why advanced oxidation processes, which are based mainly on the production of these radicals, are so efficient (see the explanation in the introduction section, lines 46-64).

Comment 5. Have you performed any supporting evidence for trapping free radicals by electron paramagnetic resonance (EPR)??

Response 5. We thank this question to the reviewer as we know that EPR is required for the unequivocal identification of hydroxyl radicals. However, our University does not have the necessary equipment. For this reason, in our previous works concerning quinone redox cycling in white rot fungi, more than one indirect procedure to detect ·OH radicals were used, such as hydroxylation of 4-hydroxybenzoic acid to 3,4-dihydroxibenzoic acid, always finding a positive correlation between the produced levels of TBARS and 3,4-dihydroxybenzoic acid (e.g., reference 33) (see lines 377-381). It was also demonstrated the formation of Fenton’s reagent when the fungi were incubated under quinone redox cycling conditions and that TBARS production was always positively correlated with the degradation of all the pollutants tested, as shown in the present study with dyes.

Comment 6. I suggest author to read and refer the following research articles are especially comparison tables will be useful to the author information’s and also UV-Visible spectroscopic studies. Heavy metal and organic dye removal via a hybrid porous hexagonal boron nitride-based magnetic aerogel, npj Clean Water, 2022, 5, 24.

Response 6. We are sorry because we think that the study referred to by the reviewer is far from being considered an advanced oxidation process that we could relate to our study. It is true that there are studies carried out with WRF of metal decontamination based on the adsorption of metals to fungal mycelium. But this is not the case of our study. 

Comment 7. I would suggest author should conduct zeta potential measurements at fixed pH, later you can explain that how your proposed sorbent materials that can interact with organic dye.

Response 7. We did not consider the zeta potential as an important parameter in this study since hydroxyl radical are produced extracellularly by the interaction of compounds in solution. In our case, the absorption of the dye into the wall of the fungus is neither a desired nor necessary process for its degradation since, as mentioned, this degradation occurs extracellularly through compounds in solution.

Comment 8. I suggest author has to provide schematic representation for mechanism of interaction with 7 type of dyes organic molecules against with the as proposed sorbent of white-rot fungus Pleurotus eryngii related materials, this will be very attractive to read this article reader, i suggest author has to provide conceptual illustration showing their interaction.

Response 8. We appreciate the reviewer's suggestion. However, we consider that traying to explain the mode of action of ·OH radicals with the wide number of dyes studied would be quite speculative and beyond the aim of our study because we have not identified the degradation intermediates produced from the attack of ·OH to the dyes. Anyway, in futures studies of advances oxidation of specific dyes, we agree with the reviewer that the elucidation of the degradative routs will be very interesting and of valuable information.

Round 2

Reviewer 1 Report

Comments and Suggestions for Authors

This manuscript can be accepted for publication

Reviewer 3 Report

Comments and Suggestions for Authors

The authors have answered all the questions and the work is ready for acceptance.